# Unmasking an Intracardiac Shunt in a Case of Persistent Unexplained Hypoxia: A Case Report

**DOI:** 10.3390/reports8010016

**Published:** 2025-01-26

**Authors:** Sanjay Sivalokanathan, Usman Saeedullah, Auston Locke, Maria Giovanna Trivieri

**Affiliations:** 1Department of Cardiology, Icahn School of Medicine at Mount Sinai Morningside, New York, NY 10025, USA; 2Department of Cardiology, Icahn School of Medicine at Mount Sinai, New York, NY 10029, USA

**Keywords:** pulmonary hypertension, intracardiac shunt, atrial septal defect, right heart catheterization, case report

## Abstract

**Background and Clinical Significance**: Pulmonary hypertension (PH) is characterized by an increase in mean pulmonary arterial pressure and pulmonary vascular resistance. It is frequently encountered in patients with significant intracardiac shunts, often necessitating the implementation of a closure device or surgical correction. Nevertheless, the occurrence of a concomitant atrial septal defect (ASD) with a right-to-left shunt inducing left ventricular dysfunction is a rare phenomenon. **Case Presentation**: A 69-year-old female patient with a history of heart failure (with preserved ejection fraction) and end-stage renal disease on hemodialysis presented to an outside facility, with syncope and hypoxia. She was recently diagnosed with severe pulmonary hypertension (measuring 86 mmHg). Right heart catheterization (RHC) revealed precapillary pulmonary hypertension (88/37/54 mmHg), prompting the initiation of intravenous epoprostenol. Nevertheless, the patient was persistently hypoxic, raising the possibility of a concomitant diagnosis. Upon review of the prior echocardiogram, which included a bubble study, an intracardiac shunt was identified. It was hypothesized that a combination of right ventricular failure and the right-to-left shunt resulting from the ASD contributed to the persistent hypoxemia. In light of this, prostacyclin therapy was continued alongside adjunctive vasopressors, resulting in clinical stabilization. The patient was eventually discharged with a treatment regimen that included subcutaneous Treprostinil. **Conclusions**: It is important to recognize that the consequences of PH are extensive, and that a rare yet significant etiology for persistent hypoxemia may be attributed to right-to-left shunting.

## 1. Introduction and Clinical Significance

Pulmonary hypertension (PH) constitutes a significant clinical disorder characterized by elevated pulmonary arterial (PA) pressure, which can lead to right ventricular failure and may be fatal, with a 5-year survival rate of 58% [1,2,3]. The etiologies of PH are diverse and are classified according to the World Health Organization (WHO) [1]. In patients with congenital heart disease, PH is typically associated with large, unrepaired shunts. Initially, there may be a benign form of high-flow hyperkinetic pulmonary hypertension; however, this condition can progressively evolve into a malignant form that induces irreversible and permanent structural alterations within the pulmonary vascular bed [4].

In such cases, the systemic-to-pulmonary shunt causes increased pulmonary blood flow, which the pulmonary vascular bed accommodates through increased capacitance. Nevertheless, sustained high pulmonary blood flow and elevated PA pressure (PAP) provoke structural changes, vascular remodeling, and endothelial dysfunction [5]. This progression ultimately results in increased pulmonary vascular resistance, which may lead to the reversal of the shunt, culminating in Eisenmenger’s syndrome. The occurrence of Group 1 PH alongside shunt reversal, attributable to an established elevation in pulmonary vascular resistance, is rare and presents both diagnostic and therapeutic challenges.

This case report outlines the circumstances of a 69-year-old patient with severe pulmonary hypertension, who exhibited right-to-left intracardiac shunting and experienced persistent hypoxia.

## 2. Case Presentation

A 69-year-old female patient with a medical history of heart failure with preserved ejection fraction, end-stage renal disease on hemodialysis and hypertension, and chronic hepatitis B presented to an outside facility with syncope and hypoxia, with an oxygen saturation level of 62%. During previous evaluations, she was diagnosed with severe pulmonary hypertension (measuring 86 mmHg). Due to recurrent episodes of hypoxia and uncertainty regarding her diagnosis, she was transferred to a quaternary care center for a comprehensive evaluation of PH. Upon admission, examination revealed an elevated jugular venous pressure (JVP) of approximately 20 mmHg, a loud second heart sound (P2), and a parasternal heave.

### 2.1. Investigations

The electrocardiogram (ECG) indicated normal sinus rhythm, with the presence of right axis deviation and left atrial enlargement (Figure 1). The pro-brain natriuretic peptide (BNP) was notably elevated at 17,000. Both rheumatological and immunological evaluations were negative. Infectious disease assessment was notable for reactivity on the Hepatitis B viral panel, while the HIV test returned negative. The chest X-ray revealed pulmonary edema with a left-sided pleural effusion (Figure 2). A computerized tomography (CT) pulmonary angiogram did not show any evidence of pulmonary emboli (PE), but it did indicate diffuse bilateral ground-glass opacities consistent with pulmonary edema, as well as enlargement of the pulmonary arteries and cardiomegaly (Figure 3). Clinical suspicion for pneumonia was deemed unlikely, as there was an absence of an elevated white blood cell count and C-reactive protein.

Transthoracic echocardiography (TTE) demonstrated preserved left ventricular function (62%), but severe right ventricular dilation with decreased function, severe tricuspid regurgitation, and significant PH (Figure 4). A ventilation/perfusion scan revealed a low probability for PE, and pulmonary function tests indicated a moderately severe restrictive ventilatory defect. Left heart catheterization revealed non-obstructive coronary artery disease. The right heart catheterization (RHC) results were indicative of precapillary PH, showing pulmonary artery pressures of 88/37/54 mmHg, a pulmonary capillary wedge pressure (PCWP) of 6 mmHg, a pulmonary vascular resistance (PVR) of 15 Wood Units (WU), and a cardiac index (CI) of 2.0 L/min/m^2^.

Based on the results of the aforementioned investigation, the patient was diagnosed with precapillary PH, defined by a mean pulmonary artery pressure (mPAP) > 20 mmHg, PCWP ≤ 15 mmHg, and PVR ≥ 2 WU.

### 2.2. Management

Epoprostenol was initiated at 2 ng/kg/min, with an initial strategy to up-titrate the dosage to approximately 30 ng/kg/min. Subsequent to the initiation of this therapy, the patient exhibited hypotension and experienced rapidly deteriorating hypoxia and pulmonary edema. A repeat RHC indicated pulmonary artery pressures of 70/28/42 mmHg, PCWP of 16 mmHg, PVR of 7 WU, and a CI of 2.4 L/min/m^2^.

In light of these findings and with a suspicion of a mixed presentation of Group I and Group II PH, the decision was made to promptly wean pulmonary vasodilators. The patient was maintained on inhaled nitric oxide and norepinephrine. However, repeated attempts to reduce vasopressor dosage and epoprostenol resulted in persistent hypotension and severe hypoxia. A multidisciplinary meeting was convened to establish the goals of care and evaluate the patient’s clinical trajectory and available therapeutic options. Review of the prior echocardiogram, performed with a bubble study, revealed an intracardiac shunt (Figure 5, Appendix A), consistent with an atrial septal defect (ASD). It was postulated that the pulmonary edema and worsening hypoxia could have been secondary to the intracardiac shunt, in conjunction with left ventricular dysfunction resulting from prolonged right ventricular failure. Consequently, a decision was made to resume titration of the epoprostenol, supported by the use of vasopressin and dopamine to sustain the mean arterial pressure above the mPAP, thereby minimizing intracardiac shunting and improving right ventricular perfusion. 

Through this treatment approach, the patient demonstrated gradual improvement and was subsequently transitioned to a low dose of dopamine and treprostinil subcutaneous therapy. The patient’s right ventricle continued to exhibit recovery, allowing for the gradual weaning of pressor medications. Prior to discharge, her pro-BNP was ~500, accompanied by improved hemodynamic parameters (PA 65/20/35, PCWP 10, PVR 5, and CI 3). At her outpatient pulmonary hypertension visit, she reported feeling significantly better and indicated a need for intermittent home oxygen therapy. Her most recent pro-BNP was 100. She remained compliant with her medication regimen and did not report any cardiovascular symptoms. Furthermore, she improved progressively in her functional class, from IV to II. Her repeat echocardiography revealed the resolution of her biventricular dysfunction (Figure 6).

## 3. Discussion

Hypoxemia in precapillary PH is associated with various factors, including ventilation/perfusion mismatch, reduced diffusion capacity, and a low mixed venous partial pressure of oxygen (PO_2_) resulting from decreased cardiac output [6]. Although it is rare, hypoxemia may also occur due to right-to-left shunting, often attributed to the reopening of a patent foramen ovale or an ASD. There is limited evidence in the medical literature regarding the diagnosis and management of this phenomenon. The incidence of hypoxemia in this context is comparable to that observed in the general population, and it may be anticipated to be as high as 30% [2,7]. However, it has rarely been discussed beyond case reports. This may be attributed to the inherent nature of unfamiliarity of the ASD/PFO or the recognition that the intracardiac shunt is a matter of significance rather than merely an incidental finding.

It may appear that individuals with an intracardiac shunt may experience better outcomes, as the offloading of the right ventricle could enhance oxygenation through a gradual increase in blood flow. However, patients with an intracardiac shunt frequently present with elevated right-sided pressures. Notably, hypoxemia serves as a potent pulmonary vasoconstrictor and may contribute to the progression of pulmonary hypertension. In addition, hypoxemia associated with right-to-left shunting exhibits minimal improvement with supplemental oxygen. More significantly, it is not an independent factor for stroke occurrence, nor associated with valvular heart disease, atrial fibrillation, and/or risk of venous thromboembolism. Therefore, treatment is aimed at the underlying condition and offloading pulmonary vascular resistance, rather than correction of the shunt [1,8]. Direct closure has been investigated, but its role remains unclear and counterintuitive.

It is challenging to recognize a concomitant intracardiac shunt in PH. The sensitivity of detecting an atrial intracardiac shunt may vary between 50% and 90% when utilizing TTE and transesophageal echocardiography, respectively. Regardless, when persistent hypoxemia is present in patients with pulmonary hypertension, it seems reasonable to exclude an intracardiac shunt.

## 4. Conclusions

PH presents a complex diagnostic challenge requiring a comprehensive understanding of its intricate and dynamic physiology. It is crucial to exclude the presence of an intracardiac shunt and recognize that right-to-left shunting is a rare consequence of PH. Therefore, a multidisciplinary approach is essential to manage this condition effectively.

## Figures and Tables

**Figure 1 reports-08-00016-f001:**
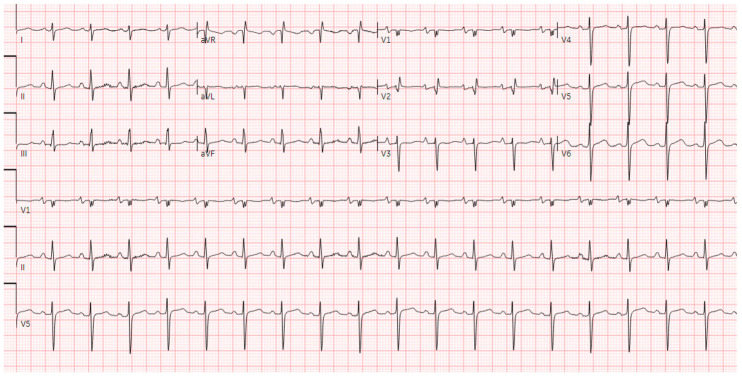
Electrocardiogram demonstrating right axis deviation and left atrial enlargement.

**Figure 2 reports-08-00016-f002:**
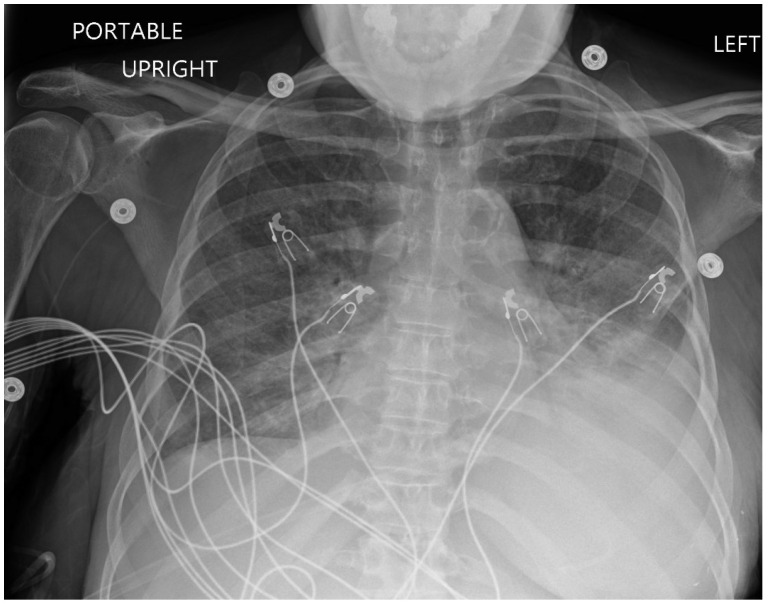
Chest X-ray showing pulmonary edema with a left-sided pleural effusion.

**Figure 3 reports-08-00016-f003:**
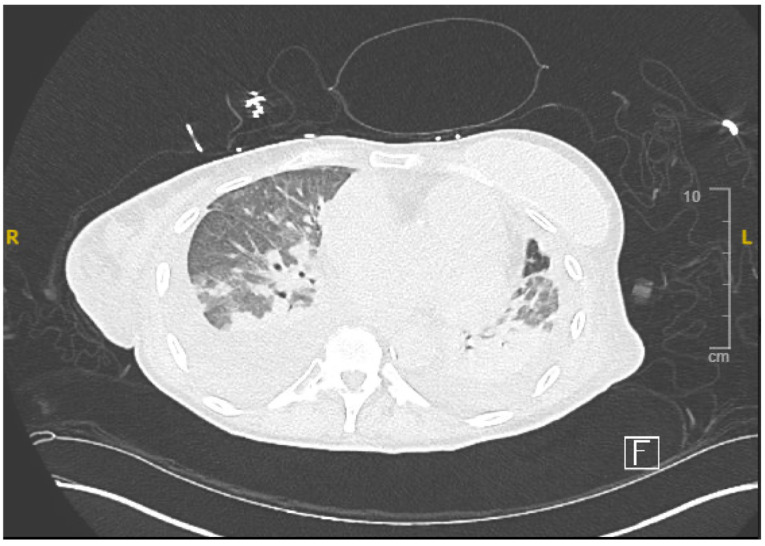
Computed tomography demonstrating bilateral ground glass opacities.

**Figure 4 reports-08-00016-f004:**
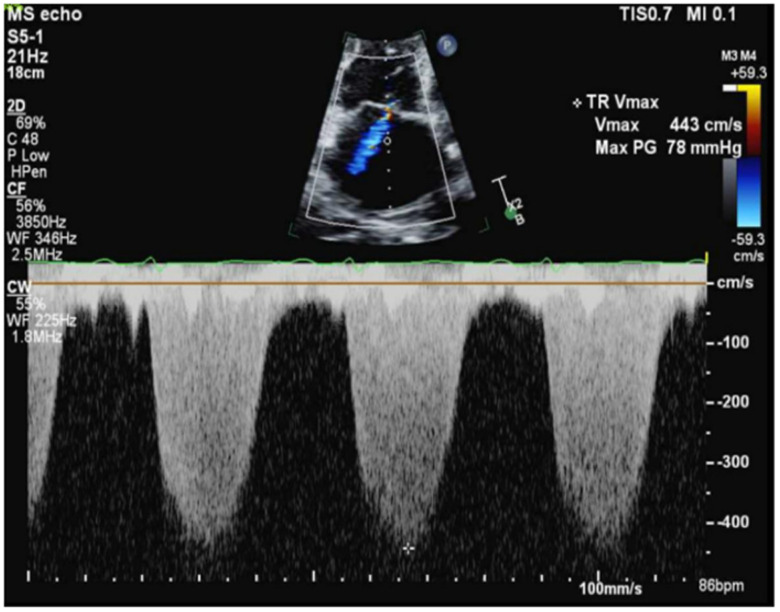
Continuous wave doppler showing severe pulmonary hypertension.

**Figure 5 reports-08-00016-f005:**
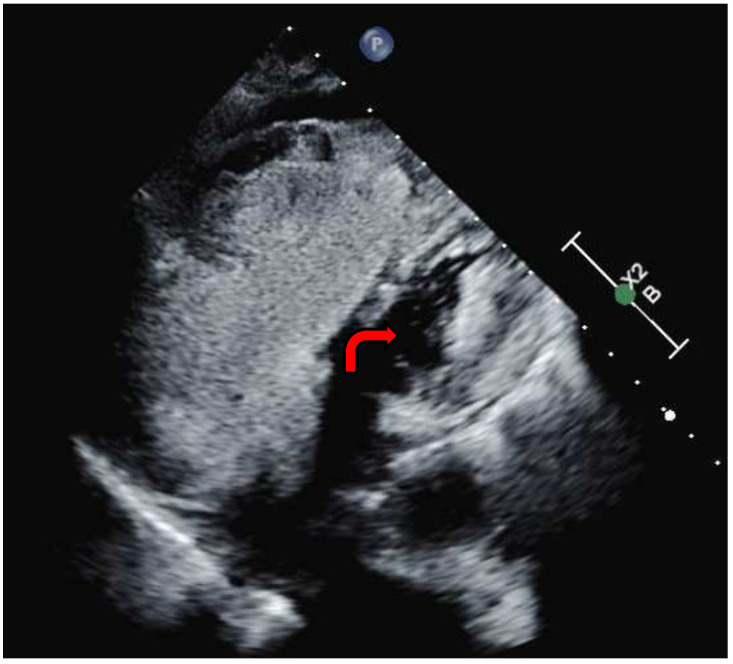
Transthoracic echocardiography with agitated saline study (bubble study). The red arrow illustrates agitated saline in the left ventricle.

**Figure 6 reports-08-00016-f006:**
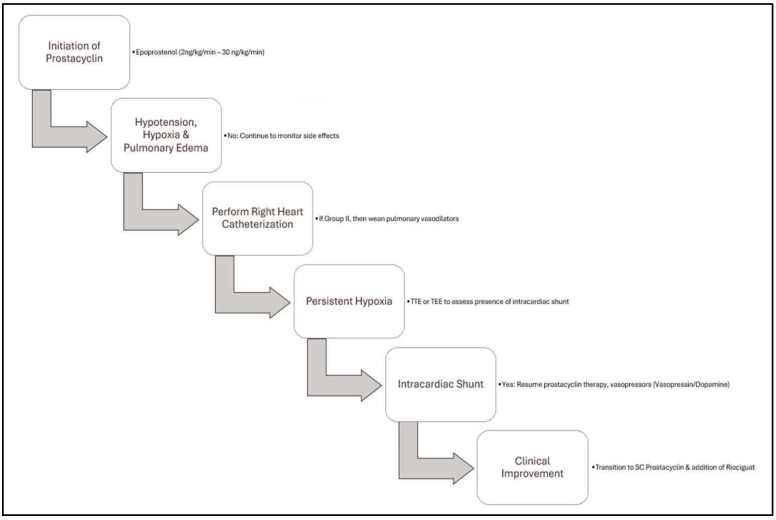
Flowchart illustrating the clinical reasoning and steps undertaken in our case.

## Data Availability

The original contributions presented in this study are included in the article/Appendix A. Further inquiries can be directed to the corresponding authors.

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
