# Peer review of "Unmasking an Intracardiac Shunt in a Case of Persistent Unexplained Hypoxia: A Case Report"

_reports, 2025, doi:10.3390/reports8010016_

Round 1
Reviewer 1 Report
Comments and Suggestions for Authors
Good presentation.
My only comment is about the CT scan, it showed significant opacities on both sides which represent atelectasis/consolidation in addition to the edema and some pleural effusion, especially on the left side. Was pneumonia/pulmonary edema excluded as a cause for her hypoxia? if yes please describe how.
Usually, biventricular heart function improves once the hypoxia resolves as the coronaries get more oxygenated blood.
Author Response
Comment 1:
My only comment is about the CT scan, it showed significant opacities on both sides which represent atelectasis/consolidation in addition to the edema and some pleural effusion, especially on the left side. Was pneumonia/pulmonary edema excluded as a cause for her hypoxia? if yes please describe how.
Reply: We thank the reviewer for this comment, and we have excluded the presence of pneumonia due to the absence of raised inflammatory markers. We have included this in our case report (Line 70-71).
Comment 2: Biventricular heart function improves once the hypoxia resolves as the coronaries get more oxygenated blood.
Reply 2: We agree with the reviewer, and the patient's biventricular function did improve, on follow-up.
Reviewer 2 Report
Comments and Suggestions for Authors
I'll provide a comprehensive review of this case report, analyzing the strengths and weaknesses of each section.
## Introduction
### Strengths
- Provides clear background information about pulmonary hypertension and its relationship with intracardiac shunts
- Effectively establishes the clinical significance of the topic
- Well-structured progression from general concepts to specific clinical scenarios
- Appropriate use of references to support key statements
- Clear identification of the knowledge gap this case report addresses
### Weaknesses
- Could benefit from more specific statistics about the prevalence of right-to-left shunts in PH patients
- The transition between general background and the specific case presentation could be smoother
- Limited discussion of current diagnostic approaches for similar cases
## Case Presentation and Methods
### Strengths
- Comprehensive presentation of patient history and clinical findings
- Logical progression of diagnostic workup
- Clear presentation of key clinical parameters and test results
- Excellent inclusion of relevant imaging (ECG, CT, echocardiography)
- Detailed documentation of hemodynamic parameters
- Well-organized chronological presentation of the case
### Weaknesses
- Some technical terms are not defined for a broader medical audience
- The sequence of diagnostic tests could be better justified
- Limited description of the decision-making process for initial treatment choices
- Could benefit from a more detailed description of the bubble study methodology
## Discussion
### Strengths
- Clear explanation of the pathophysiology of hypoxemia in precapillary PH
- Good integration of the case findings with existing literature
- Appropriate acknowledgment of the rarity of the condition
- Logical connection between clinical findings and therapeutic decisions
### Weaknesses
- Relatively brief discussion section given the complexity of the case
- Limited comparison with similar cases from literature
- Could include more detailed discussion of alternative treatment approaches
- Missing discussion of potential long-term outcomes and prognosis
- Limited exploration of the implications for clinical practice
## Conclusions
### Strengths
- Concise summary of key learning points
- Emphasizes the importance of multidisciplinary approach
- Highlights the significance of considering intracardiac shunts in PH
### Weaknesses
- Could provide more specific recommendations for clinical practice
- Limited discussion of future research directions
- Missing suggestions for preventive strategies or early detection
- Could benefit from more detailed follow-up recommendations
## Overall Assessment
This case report presents a clinically significant and well-documented case of pulmonary hypertension complicated by an intracardiac shunt. The manuscript's primary strength lies in its detailed clinical presentation and comprehensive diagnostic workup. The authors effectively demonstrate the importance of considering intracardiac shunts in cases of unexplained hypoxia in pulmonary hypertension patients.
However, the discussion section could be expanded to provide more context and practical implications. The conclusions, while accurate, could offer more specific guidance for clinicians encountering similar cases.
### Recommendations for Improvement
1. Expand the discussion section to include:
- More comparative cases from literature
- Detailed analysis of treatment decisions
- Long-term management strategies
- Potential complications and their management
2. Enhance the conclusions by adding:
- Specific clinical recommendations
- Follow-up protocols
- Risk stratification guidance
- Research gaps and future directions
3. Consider adding a dedicated section on:
- Diagnostic algorithms for similar cases
- Treatment decision frameworks
- Prevention strategies
- Quality of life considerations
Despite these limitations, the case report makes a valuable contribution to the literature on pulmonary hypertension and intracardiac shunts, providing important insights for clinicians managing similar cases.
Possible addition to the references:
I would recommend citing this paper about PH and Covid for the following reasons:
DOI: 10.5603/CJ.a2021.0159
1. Related Clinical Presentations:
- Both papers deal with pulmonary hypertension and right ventricular dysfunction
- Both discuss cases of unexplained hypoxia, though through different mechanisms
- Both emphasize the importance of thorough cardiac evaluation in cases of persistent hypoxia
2. Diagnostic Value:
- The COVID paper provides additional context about right heart catheterization findings in patients with pulmonary vascular complications
- It offers comparative hemodynamic parameters that could serve as reference values
- Both papers emphasize the importance of comprehensive cardiac evaluation including right heart catheterization
3. Pathophysiological Insights:
- The COVID paper explains mechanisms of right ventricular adaptation to increased pulmonary vascular resistance, which is relevant to understanding the shunt physiology in the first paper
- It provides additional context about how right ventricular dysfunction develops in response to increased pulmonary pressures
4. Clinical Relevance:
- Both papers discuss the importance of recognizing and managing right ventricular dysfunction
- The COVID paper could provide additional context about how viral infections can lead to pulmonary vascular complications
- Both emphasize the importance of follow-up and monitoring of pulmonary pressures
5. Treatment Considerations:
- Both papers discuss the management of patients with right ventricular dysfunction and pulmonary hypertension
- They both demonstrate the importance of serial evaluations and monitoring of clinical improvement
This paper could be cited in several sections of the original case report:
- In the introduction when discussing causes of pulmonary hypertension
- In the discussion section when explaining mechanisms of right ventricular dysfunction
- In the sections discussing diagnostic approaches to unexplained hypoxia
- When discussing the importance of follow-up and monitoring
The COVID paper would add valuable context about how viral infections can affect pulmonary vasculature and right heart function, which could broaden the discussion of the case report's findings.
Author Response
Dear Reviewer,
We sincerely appreciate your insightful comments, which have significantly enhanced the robustness of our case report. In response, we have incorporated a comprehensive discussion that addresses additional cases, treatment options, and includes a flowchart illustrating our management approach for this patient. Furthermore, we have added the reference article you provided.
Kind regards,
Sanjay Sivalokanathan
Reviewer 3 Report
Comments and Suggestions for Authors
Here’s the translated text into professional medical English:
"First of all, I would like to thank the editor for the opportunity to revise this manuscript. The manuscript presents a case study of a patient with an undetected atrial septal defect (ASD), which manifested only as desaturation during a right-to-left shunt. The work is succinct and clearly written. Despite the fact that the presented case is not particularly rare, the study effectively documents the successful conservative therapy of a critical clinical condition.
I have only one comment regarding the work: The authors state in the text that the X-ray image documented a left-sided pleural effusion. However, the authors do not provide corresponding radiological documentation for this finding. The attached CT image, however, documents bilateral pleural effusion, with the effusion on the right side being significantly more pronounced. This correlates with the typical clinical course, wherein a right-sided pleural effusion is a larger and earlier sign of decompensation compared to its left-sided presence. I kindly request that the authors provide documentation for the X-ray image indicating the left-sided pleural effusion.
Upon addressing my requests, I recommend the manuscript for publication."
Author Response
Dear Reviewer,
Comment 1: I have only one comment regarding the work: The authors state in the text that the X-ray image documented a left-sided pleural effusion. However, the authors do not provide corresponding radiological documentation for this finding. The attached CT image, however, documents bilateral pleural effusion, with the effusion on the right side being significantly more pronounced. This correlates with the typical clinical course, wherein a right-sided pleural effusion is a larger and earlier sign of decompensation compared to its left-sided presence. I kindly request that the authors provide documentation for the X-ray image indicating the left-sided pleural effusion.
Reply: We appreciate the reviewer for their observation. In response, we have incorporated the Chest X-ray into our case report. Additionally, we have added a statement indicating that the patient did not exhibit any elevated inflammatory markers that would suggest the presence of pneumonia.
Warm regards,
Sanjay Sivalokanathan
Round 2
Reviewer 1 Report
Comments and Suggestions for Authors
Looks Good.
Reviewer 3 Report
Comments and Suggestions for Authors
Thank you to the authors for providing the requested documentation.
I recommend publishing the manuscript in its current form.